# An Overview of Peptide-Based Molecules as Potential Drug Candidates for Multiple Sclerosis

**DOI:** 10.3390/molecules26175227

**Published:** 2021-08-28

**Authors:** Annarita Del Gatto, Michele Saviano, Laura Zaccaro

**Affiliations:** 1Institute of Biostructures and Bioimaging, CNR, 80134 Naples, Italy; 2CIRPeB, University of Naples “Federico II”, 80134 Naples, Italy; msaviano@unina.it; 3Institute of Crystallography, CNR, 70126 Bari, Italy

**Keywords:** demyelinating diseases, multiple sclerosis, peptides, therapy, EAE

## Abstract

Multiple sclerosis (MS) belongs to demyelinating diseases, which are progressive and highly debilitating pathologies that imply a high burden both on individual patients and on society. Currently, several treatment strategies differ in the route of administration, adverse events, and possible risks. Side effects associated with multiple sclerosis medications range from mild symptoms, such as flu-like or irritation at the injection site, to serious ones, such as progressive multifocal leukoencephalopathy and other life-threatening events. Moreover, the agents so far available have proved incapable of fully preventing disease progression, mostly during the phases that consist of continuous, accumulating disability. Thus, new treatment strategies, able to halt or even reverse disease progression and specific for targeting solely the pathways that contribute to the disease pathogenesis, are highly desirable. Here, we provide an overview of the recent literature about peptide-based systems tested on experimental autoimmune encephalitis (EAE) models. Since peptides are considered a unique therapeutic niche and important elements in the pharmaceutical landscape, they could open up new therapeutic opportunities for the treatment of MS.

## 1. Introduction

Multiple sclerosis (MS) is a chronic inflammatory disease of the central nervous system (CNS) that leads to progressive neurodegeneration. A crux of the disease is an autoimmune attack of the self-antigens, that is the proteins of the myelin sheath that wrap around the nerve fibers are mistaken for foreign agents by macrophages, CD4^+^T, CD8^+^T, and B cells infiltrating the blood-brain barrier (BBB). The role of the myelin sheath is crucial both to provide protection and nourishment to nerve fibers and to enable the efficient transmission of nerve impulses. Demyelination elicits axonal injury, plaques, or lesions’ formation in the brain and spinal cord, and causes a range of symptoms, depending on the neurons that are affected [1,2,3,4,5]. MS is a very heterogeneous disease, with different pathological and clinical manifestations. It is estimated that worldwide, more than 2.3 million people between ages 20 to 50 are affected by MS, that is usually diagnosed with signs and symptoms that may differ from person to person and throughout the disease, such as loss of balance and coordination, visual and sensory deficiency, fatigue, weakness, vertigo, pain, and cognitive difficulties [6]. Before being officially diagnosed with MS, patients can experience a first neurologic event indicative of potential MS, defined as Clinically Isolated Syndrome, that lasts for at least 24 h with symptoms and signs indicating either a single lesion (monofocal) or more than one lesion (multifocal) within the CNS. Multiple sclerosis is divided into four types, named according to the way the disease acts on the body over time: Progressive Relapsing MS (PRMS), characterized by a steady degeneration since onset with super-imposed attacks, Relapsing-Remitting MS (RRMS), the most common disease course diagnosed for 85% of people, characterized by defined attacks of new or even increasing neurological symptoms (relapses), followed by periods of partial or complete recovery (remissions), Primary Progressive MS (PPMS), consisting of progressive disability from the onset of symptoms, and Secondary Progressive MS (SPMS), marked by an initial relapsing-remitting course that suddenly begins to progressively decline over time [7,8].

### 1.1. Immunopathogenesis of MS

The pathogenesis of MS is still unknown: immune-system cells and proteins that can have a crucial role in the initiation and progression of disease are not fully understood. A multitude of factors may cause the insurgence of this disease, ranging from genetic factors, such as class-II allele of the major histocompatibility complex (MHC) [9,10], to environmental agents, such as pathogenic infections [11,12]. Some correlations actually have been made between MS and different viruses, such as varicella zoster [13] and the Epstein-Barr virus [14,15], or bacteria such as chlamydia pneumonia [16,17,18], as well as other environmental factors such as vitamin D deficiency, smoking, and obesity [8]. Despite this, direct evidence of a correlation between pathogenic infections and MS is still lacking, with the most commonly accepted hypothesis being that MS is an autoimmune disease affecting genetically predisposed individuals already plagued by an environmental pathogen [19].

An intricate interplay of inflammatory cells (T cells, B cells, and macrophages) and CNS resident cells (microglia, astrocytes, oligodendrocytes, and neurons) is associated with the pathology. Among the different cells of the innate and adaptive immune systems that may arrange the inflammatory response within the CNS, a key role is played by autoreactive CD4^+^T cells that migrate into the CNS [20,21,22,23,24] after the activation in the peripheral lymph nodes. Many findings indicate a central role of Th1, Th17, Th1–Th17-like, Th22, and Granulocyte-Macrophage Colony-Stimulating Factor (GM-CSF)-producing CD4^+^T cells, in beginning and prolonging the inflammatory reactions and in causing neurodegeneration in MS as well as dysfunction in regulatory T cell (T_reg_) subsets [25,26]. In the CNS, these cells are reactivated and secrete cytokines and chemokines, thus contributing to the breakdown of the BBB, the activation of resident astrocytes and microglia, and finally, the mediation of inflammatory reaction responsible for the distinctive lesions of the disease [27].

### 1.2. Current Available Therapeutic Approaches

At present, there is no definite cure for MS, but therapies based on immunosuppressive and immune-modulating agents can modify or slow the course of disease and manage symptoms. Several disease-modifying treatments (DMTs) to treat relapsing forms of MS have been developed and proved able to reduce the frequency and severity of relapses as well as the accumulation of lesions in the CNS. They cover injectable and orally administrated drugs, monoclonal antibodies, transplantation of autologous bone marrow, strategies to restore myelination, and targeting T cells, besides mesenchymal stem cells’ employment. So far, about 20 DMTs, approved by the FDA, are available in many countries for the different types of MS (www.nationalmssociety.org/For-Professionals/Clinical-Care/Managing-MS/Disease-Modification, accessed on 3 August 2021), but they can only minimize symptoms, preserving patients as relapse-free, with no new lesion visible on MRI. Moreover, the most current therapies are not specific but suppress the general immune response, leading to many negative side effects. For this, the development of therapeutic agents able to specifically control and modulate the immune response is an urgent need.

In this context, peptide molecules appear as appealing candidates for many reasons [28]. In fact, some of them have a deep impact on important physiological mechanisms controlling many vital functions in humans, such as metabolism, respiration, reproduction, and immune defense. They possess many advantages over other classes of molecules, being small in size (up to 50 amino acids), easy to synthesize, and having the ability to penetrate the cell membranes, mainly when bearing specific modifications (N-methylation, stapling, etc.). They also have high activity, specificity, and affinity, good efficacy and tolerability, as well as biological and chemical diversity. As a result, an increasing number of peptides are both entering clinical trials and being approved as drugs.

In this review, attention is pointed to some peptides reported in the recent literature representing new potential candidates for MS therapy. Generally, for pre-clinical studies, the experimental autoimmune encephalitis is the most commonly used immune-driven demyelinating model that closely resembles RRMS. It is an animal model of brain inflammation induced by immunization with neural antigens, such as myelin oligodendrocyte glycoprotein (MOG) or proteolipid protein (PLP). Additionally, the cuprizone model, that mimics the acute and chronic courses of MS and is thus suited for the study of the demyelination mechanism, represents a valid tool to develop novel therapies aimed at protecting oligodendrocytes and promoting remyelination.

Here, we describe peptide systems arbitrarily sorted into two groups: peptides modulating T cells’ functions and peptides modulating cells other than T lymphocytes (Figure 1).

## 2. Peptides Modulating T Cells’ Functions

Currently, MS therapies are unable to specifically regulate immune cells because they suppress the overall immune system, causing undesirable side effects from opportunistic infections. Thus, to preserve host capability provided by the general immune response to protect against extraneous pathogens, the development of therapeutic agents capable of specifically controlling the myelin-reactive immune response is required.

### 2.1. Myelin-Based Autoantigen Peptides: ATX-MS-1467, c_(91–99)_[A^96^]MBP_87–99_, c_(87–99)_[A^91,96^]MBP_87–99_, and c_(87–99)_[ R^91^, A^96^]MBP_87–99_

Antigen-specific treatments targeting key regulators in the failure of tolerance to self-antigens, such as myelin basic protein (MBP), were developed to silence or reprogram autoreactive T cells in the peripheral zone to a regulatory phenotype, thus creating a tolerogenic state to the targeted protein [29,30]. Autoantigen T-cell responses to MBP are known to be involved in the pathogenesis of MS [31,32,33,34], and, in particular, five of the eight identified regions of MBP are the most recognized by T-cells. Based on this consideration, Streeter and coworkers in 2015 [35] reported for the first time a study on ATX-MS-1467, a cocktail of four peptides derived from the MBP region (MBP 30–44 [ATX-MS1], MBP 130–144 [ATX-MS4], 140–154 [ATX-MS6], and MBP 83–99 [ATX-MS7], Table 1), which resulted acting as apitopes (antigen processing independent epitope) and to prevent the worsening of signs of disease in EAE in a humanized mouse, (DR2 9 Ob1)F1, in a dose-dependent fashion. A few years later, De Souza et al. [36] demonstrated that subcutaneous treatment with an ATX-MS-1467 mixture, after established induction of EAE in the same mouse model, was able to reverse the clinical disability, to diminish the histological markers of inflammation and demyelination, and to reduce the T-cell and B-cell infiltration in the spinal cord and the disruption of BBB integrity, as shown in MRI analysis. All these effects were more evident in the animals treated early in the progression of EAE rather than later, thus demonstrating the neuroprotective benefit of the ATX-MS-1467 mixture when early administration was carried out. Furthermore, De Souza et al. speculated that chronic treatment with ATX-MS-1467 induced a shift from a pro-inflammatory to a tolerogenic state in the periphery, as revealed by the increase in IL-10 secretion relative to cytokine IL-2, IL-17, and IFN-c, and by the induction of splenic-induced T_regs_ as resulting from a decrease in splenocyte proliferation and the increase in IL-10^+^Foxp3^−^ T regulatory cell sub-populations in the spleen. ATX-MS-1467 is in clinical Phase II in patients with RRMS to evaluate the clinical, biological, and radiological effects and the safety profile (https://www.mstrust.org.uk/a-z/atx-ms-1467, accessed on 3 August 2021).

Different cyclic peptides derived from MBP_87–99_ epitope [37,38,39,40,41,42], another candidate autoantigen in MS, were also explored for therapeutic efficacy in EAE. Previous studies [43,44] evidenced the importance of Lys91 and Pro96 of MBP_87–99_ in anchoring T-cell receptor (TCR) and in the involvement in the trimolecular complex between the TCR–ntigen (peptide)–MHC for the activation of encephalitogenic T cells necessary for EAE induction (Table 1). In fact, the replacement of these two residues with Arg and/or Ala, respectively, induced protective effects against EAE in mice. The peptide analogues cyclo (91–99) [Ala96] MBP_87–99_, cyclo (87–99) [Ala91,96] MBP_87–99_, and cyclo (87–99) [Arg91, Ala96] MBP_87–99_ (Table 1) were administered in MBP_72–85_-induced EAE in Lewis rats using prophylactic and early therapeutic vaccination protocols [45]. These peptides, but not wild-type linear MBP_87–99_, strongly inhibited EAE, and in particular, cyclo (87–99) [Arg91, Ala96] MBP_87–99_ was highly active in preventing the onset and development of clinical symptoms and pathologies correlated with the spinal cord, providing long-term protection against EAE induction through an active mechanism.

### 2.2. Peptides Modulating Th Cell Subsets: Cilengitide, TnP Peptide, [T20K] Kalata B1, Thymulin

It is widely documented that integrins are transmembrane receptors promoting the migration of cells into inflamed tissues by interactions with inflamed endothelium and stromal extracellular matrix constituents. Hence, molecules able to interfere with, or even to block, integrin activity can be employed as therapeutics in the treatments of MS. An example is the monoclonal antibody Natalizumab, approved by the FDA for the treatment of RRMS, which acts as an α4 integrin antagonist to prevent migration of inflammatory T cells into the CNS. The antibody proved highly effective in some patients, even if its administration is associated with the risk of progressive multifocal leukoencephalopathy. Lately, it has been reported that a4 integrin is definitively not essential for cell access of Th17 to the CNS. Recent data pointed out the key role of interleukin 23 (IL-23) in triggering the pathogenic features in pro-inflammatory Th17 in EAE by inducing Th17 cell proliferation, and particularly the switch to effector phenotype after the initial signals for differentiation induced by transforming β growth factor, IL-6, and IL-1 [46,47,48,49]. As evidence of this, mice lacking in IL-23 or IL-23R were highly resistant to Th17-mediated autoimmune inflammation. In the last years, different groups independently stated an increased integrin β3 expression in Th17-associated diseases, such as psoriasis, psoriatic arthritis, rheumatoid arthritis, and MS [50,51,52,53]. In addition, this integrin is known to bind ECM proteins, such as osteopontin, tightly associated with autoimmune diseases [54], and vitronectin and fibronectin, for which increased levels of expression in the CNS in both EAE and MS have been found [55,56]. In 2016, Du and collaborators reported for the first time [57] a deep study on the expression and function of αvβ3 integrin on Th17 cells in the context of autoimmune disease in EAE model and elucidated its interesting connections with the IL-23/Th17 axis. Notably, an increased integrin αvβ3 expression in an IL-23-dependent manner was observed only on Th17 lineage, that expanded during the progression of EAE, but not on all activated T cells or Th1 cells. These findings supported a further role of the integrin in promoting migration of inflammatory T cells into the CNS parenchyma to start inflammation. By confirming these data, the cilengitide (Table 1), an αvβ3/αvβ5 antagonist, was administered from day 4 after animal immunization with MOG (35–55) and a significant reduction of EAE severity over untreated mice was observed. These results indicate a crucial targetable function for integrin αvβ3 expression by inflammatory Th17 cells and pave the way for testing further selective αvβ3 antagonists [58,59] in the EAE model as a different perspective in the MS scenario.

Komegae et al. [60] identified new peptide molecules, denominated the TnP family, derived from the venom of *Thalassophryne nattereri* Brazilian fish, which was utilized for drug discovery and development. The TnP family, subjected to a patent application in several countries all over the world, comprises synthetic peptides with anti-inflammatory and anti-allergic activities. In particular, the authors focused on a specific TnP peptide (Table 1) and employed the myelin-dependent EAE model [61] to evaluate its anti-inflammatory effect and therapeutic potential in MS. Moreover, they used the toxic model of cuprizone [62,63] to also assess its potential to induce remyelination. They reported very interesting results: treatment with TnP peptide enhanced EAE in an IL-10-dependent manner, prompting reduction of disease severity and the beginning of symptoms. A controlled infiltration of leukocytes in CNS, demyelination inhibition, decrease in the expansion of microglia and of the activity of MMP-9 by F4/80^+^ macrophages were also evidenced. TnP proved able to modulate the encephalitogenic CD4^+^ T cells, reducing the infiltration of IFN-γ-producing Th1 and IL-17A-producing Th17 cells in the CNS, to block inflammatory cytokines’ production in the spleen, and to promote the presence of functional Treg in the spleen and the CNS. Finally, the TnP peptide showed the ability to accelerate remyelination in the cuprizone model, proving to be a very active and interesting anti-inflammatory and pro-remyelinating molecule to use as a lead compound to design new drugs for demyelinating conditions typical of MS.

Many drugs for MS treatment are unfortunately administered by the parenteral route, and innovative orally active therapeutics are desirable, mainly for the treatment of chronic forms. [T20K] kalata B1 (Table 1) is a plant-derived peptide belonging to the family of cyclotides, well-known for their cyclic fold stabilized by a cystine knot motif of three disulfide bonds resistant to chemical, enzymatic, and thermal degradation [64]. As a result, cyclotides are very attractive weapons in drug discovery and development because they may be administered by the oral route.

An in vitro effect of [T20K] kalata B1 in inhibiting T-cell proliferation through the decrease of the expression of the IL-2 surface receptor as well as IL-2 cytokine secretion and IL-2 gene expression was reported [65]. This is extremely relevant considering the physiological role of IL-2 in T-lymphocyte activation and its action as an autocrine factor for stimulating T-cell proliferation, with consequent loss of immune tolerance. Recently, Thell and coworkers thoroughly described the ability of the peptide to reduce IL-2 and IFN-γ secretion as well as to significantly lower the degree of demyelination and inflammation in EAE mice after oral treatment [66]. Notably, a considerable decrease in T-cell cytokine IL-17A, supporting the observed clinical and histological reversion of disease evolution upon peptide treatment, was also evidenced. Remarkably, this study for the first time evidences the use of cyclotides as potential oral active therapeutics for EAE.

NF-κB is one of the most important transcription factors, whose activation in inflammatory cells, including T cells, and some resident cells such as microglia/macrophages and astrocytes, increases inflammation and promotes the development of MS and EAE [67]. There is also abundant evidence that the NF-κB pathway is crucial for maintaining immunological tolerance as it participates in the negative selection of autoreactive T cells and the selection and maintenance of Treg cells. Lunin et al. [68] showed that the induction and maintenance of acute EAE forms were associated with NF-κB pathway activation, and NF-κB signaling inhibition alleviated acute EAE symptoms. The regulation of the NF-kB pathway in EAE is attained by site-specific phosphorylation of a RelA/p65 protein from the NF-kB family [68]. An increased RelA/p65 (Ser276) phosphorylation, commonly attributed to PKA, corresponded with the production of IF-γ, Hsp72, and the initial phase of IL-17 production associated with the early phase of the disease, whereas increased RelA/p65 (Ser536) phosphorylation was correlated not only with the activation of IKK, SAPK/JNK, and p53, but also with the late phase of IL-17 secretion and the pathology progression. Lunin and coworkers [69,70] experienced that the thymic peptide thymulin, a metallopeptide consisting of a nonapeptide (Table 1) in complex with a zinc ion, in addition to producing effects on the neuroendocrine stress response system [69,70,71], shows analgesic activity in inflammatory conditions [72] and in severe and mild progressive forms of EAE [73,74]. Recently [75], the thymulin was bound to polybutylcyanoacrylate (PBCA) nanoparticles for enhancing the bioavailability of the free peptide [76], and the properties of both forms of the peptide were also evaluated on relapse and remitting EAE. It was demonstrated that the ability to reduce the phosphorylation at Ser536 was the same for thymulin and PBCA-bound thymulin, while a deeper effect on phosphorylation at Ser276 was observed in animals treated with peptide-conjugate with respect to free peptide. From this evidence, the PBCA-bound thymulin was also found to be more active in reducing overall symptoms of the disease in comparison with free thymulin.

## 3. Peptides Modulating Other Cells besides T Lymphocytes

Since there is such a large number of MS immunomodulatory therapies with different modes of action already available, many experts believe that the research aiming to develop truly novel immune therapies is unsuccessful. Despite this huge choice of treatments, none are capable of fully silencing the pathology and failures occur with all the accessible drugs. Thus, MS treatment needs to advance with the development of neuroprotective and regenerative therapies aiming to complement existing immunomodulatory therapies [77].

### 3.1. Remyelination-Enhancing Peptide: TDP6

For remyelination, it is necessary to have a presence of functionally undamaged axons and oligodendroglial progenitor cells (OPCs) that can differentiate, form contact with and ensheath axons, as well as generate compact myelin [77]. Brain-derived neurotrophic factor (BDNF, Table 1) plays extremely important roles in CNS myelination, acting by oligodendrocyte-expressed TrkB receptors. BDNF exerted an effect in increasing myelin sheath thickness, but the failure of the clinical trials in neurodegenerative diseases was due to its promiscuity to interact with both p75^NTR^ and TrkB receptors, and therefore its short half-life, large molecular size, and low capability to cross the BBB [78]. A recent paper by Fletcher et al. [78] reported the tricyclic-dimeric-peptide-6 (TDP6, Table 1), a BDNF mimetic, that selectively interacts with the oligodendroglial TrKB receptor and not with the p75^NTR^ one. The peptide enhances remyelination, increases oligodendrocyte differentiation, the frequency of myelinated axons, and the myelin sheath thickness following a demyelinating insult. All these outcomes support the modulation of TrkB signaling as an effective therapeutic strategy to further myelin repair.

### 3.2. Peptide Inhibiting Microglia Activation: γ^377–395^

Significant data indicate that in MS lesions, a co-localization of perivascular activated microglia with areas of BBB breakdown occurs [79]. One of the main events correlated with the BBB disruption is the escape of blood protein fibrinogen in the CNS and relative perivascular deposition of fibrin that interacts with the Mac-1 microglia receptor. This action induces local activation of microglia and increases its capability to phagocytose myelin and secrete pro-inflammatory cytokines. Adams and colleagues reported that the fibrinogen-derived peptide γ^377–395^ (Table 1) can directly block fibrinogen-Mac-1-dependent microglial activation both in vitro and in vivo without effects on coagulation [80]. Specifically, the peptide attenuates the clinical symptoms in EAE by mainly reducing the microglia/macrophage response without affecting T cell activation. Accordingly, the inhibition of fibrin-Mac-1 interactions could be helpful as a microglia-suppressive therapy in MS and could be employed in association with other therapies targeting diverse systems involved in the onset of the disease.

### 3.3. Peptide Regulating T Cell Subsets and CNS Resident Cells: JM4

Beneficial immunomodulatory and anti-inflammatory effects of Erythropoietin (Epo) have been demonstrated in the MOG-Induced Monophasic EAE model [81]. Unfortunately, these features are associated with excessive hematopoiesis that prevents Epo application in the therapeutic field [82,83,84]. In 2015, Yuan’s group designed a peptide derived from Epo protein, named JM4 (Table 1), which shares the valuable properties of whole molecule Epo in EAE mice, but not its side effects [85,86,87]. The authors demonstrated that both Epo and JM4 act by reducing raised mononuclear cell counts to normal, dendritic cells by ten-fold, and proinflammatory cytokines such as IL-2, IL-6, TNF-alpha, and INF-gamma. Moreover, JM4 peptide can normalize the damaged BBB, downregulate the MHC class II expression, enhance the expansion of the Treg-cell population, and reduce T helper Th17-positive cells in SJL/J relapsing-remitting EAE mice [85]. Notably, the peptide also sustains the neuroprotective A2 astrocytes over the cytotoxic A1 astrocytes, the latter of which are upregulated in MS, where they trigger the death of oligodendrocytes and neurons [88]. Additionally, since microglial activation is noted to induce A1 reactive astrocytosis, JM4 may act by suppressing the M1 macrophages or microglia development. Importantly, the long-term therapeutic benefit with no side effects observed with a short-term JM4 treatment in EAE mice [89] looks like that recently described in patients treated with the pulsed immune reconstitution therapy, alemtuzumab and cladribine, that are the new FDA-approved drugs for MS therapy [90].
molecules-26-05227-t001_Table 1Table 1Peptide sequences.NameSequenceRef**ATX-MS-1467**ATX-MS1: PRHRDTGILDSIGRF[34]
[ATX-MS4]: GFKGVDAQGTLSKIF

[ATX-MS6]: GFKGVDAQGTLSKIF

[ATX-MS7]: ENPVVHFFKNIVTPRTP
**cyclo (91–99) [Ala^96^] MBP_87__–__99_**
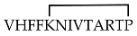
[44]**cyclo (87–99) [Ala^91,96^] MBP_87__–__99_**
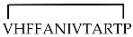

**cyclo (87–99) [Arg^91^,Ala^96^] MBP_87__–__99_**
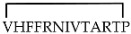

**Cilengitide**c(RGDf(NMe)V)[56]**TnP peptide**
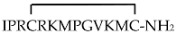
[59]**[T20K]kalata B1**
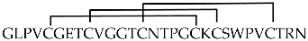
[65]**Thymulin**EAKSQGGSD[74,75]**TDP6**
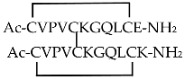
[77]**γ377-395**YSMKKTTMKIIPFNRLTIG [79]**JM4**GCAEHCSLNENITVPDTKV[88]

## 4. Conclusions

Multiple sclerosis is a serious neuroinflammatory disease with disseminated focal lesions affecting myelin and neuronal loss in the CNS, clinically characterized by unpredictable course and increasing disability. To date, 22 DMTs approved by the FDA are available in the clinic for the different types of MS. Unfortunately, the efficacy, tolerability, and safety profile of these drugs vary widely, ranging from modest activity with remarkable safety to highly effective activity with increased risk of serious adverse events also associated with the unspecific mechanisms of action. Consequently, the discovery of new therapeutic agents acting by specific control and modulation of the immune response is a great challenge.

Over the last decades, the use of peptides as drugs has advanced and continues to grow with changes in drug development and treatment protocols.

An illustrative but not exhaustive list of different peptides as potential candidates in MS therapy has been reported here. The use of these molecules offers numerous advantages, even if in some cases the low solubility and enzymatic stability or the suboptimal pharmacokinetic properties may limit their employment. These drawbacks could be overcome by introducing chemical modifications in peptide backbone, or directly conjugating these peptides with adequate vehicles (peptides, liposomes, nanoparticles) to improve targeted delivery also through the BBB, and/or therapeutic activity. In addition, the preparation of dendrimers made up of multiple copies of the same or different peptides in a unique molecule provides branched systems with synergistic effects and almost unlimited potential in biomedicine.

In conclusion, the interesting results and the perspectives reported here open a new scenario for drug innovation in the MS field.

## Figures and Tables

**Figure 1 molecules-26-05227-f001:**
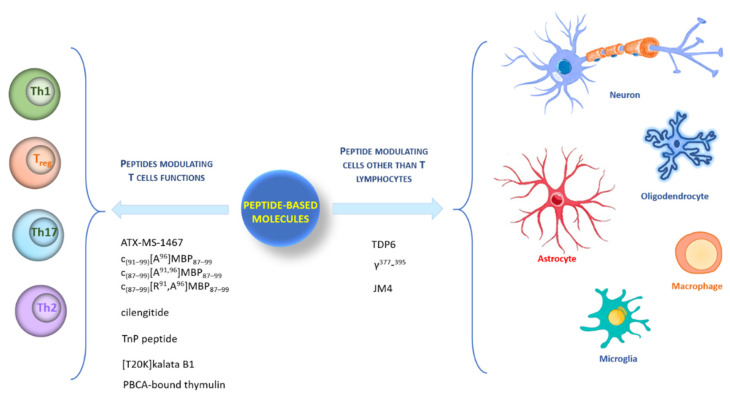
Graphic representation of different type of cells targeted by peptides.

## Data Availability

Not applicable.

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
