# Peer review of "An Overview of Peptide-Based Molecules as Potential Drug Candidates for Multiple Sclerosis"

_molecules, 2021, doi:10.3390/molecules26175227_

Round 1

Reviewer 1 Report

Multiple sclerosis (MS) is an autoimmune disease that attacks the central nervous system (CNS) causing progressive neurodegeneration.

Demyelination a whole mark of MS, elicits axonal injury, plaques, or lesions formation, in the brain and spinal cord, causing a range of symptoms. Current therapies are based on immuno-suppressive and immune-modulating agents. Several promising disease-modifying treatments (DMTs) to treat relapsing forms of MS have been developed.

In this manuscript, the authors provide an overview of the recent literature about peptide-based systems tested on experimental autoimmune encephalitis models that use neural antigens such as myelin oligodendrocyte glycoprotein or proteolipid protein. These represent a unique therapeutic modality and could open up new therapeutic opportunities for the treatment of MS.

The manuscript is well written and summarizes the pertinent information. A good read for the novice in this area and an update for the seasoned investigators.

Author Response

The authors are thankful for the positive comments.

Reviewer 2 Report

In this paper the authors describe peptide therapeutics for the treatment of multiple sclerosis. The review appears to be scientifically sound and of interest in the field since there is little literature on peptide therapeutics used in MS.

In spite of this, several aspects should be improved:

The review is quite heavy to read and difficult to follow in its current form. A clearer structure is required. Subtitles in each section would greatly help the reader.

This review reads as a listing of peptides related to MS. Some perspective and trend in the kind of peptides used missing.

A reference to a recent comprehensive review on the topic by Rayatpou and Javan (Pharmacological Research, 2021, 167, 105441) is missing.

Acronyms in the abstract should be explained (MS and EAE).

EAE should be better defined and linked with MS.

Some explanations should be improved. For instance, the second sentence in the Introduction is confusing and the fragment “macrophages infiltrated the blood-brain barrier (BBB)” is inconsistent.

In some instances, more formal expressions should be sought. For instance, in page 2, “Till now,” in the third paragraph” should read “Until now,”.

Peptides can be outstanding therapeutics but not all of them have the properties described in the las paragraph in page 2. Please nuance these properties and state that peptides “may have” them. For instance, it is said peptides have a “small size”, how small? Actually, only a small fraction of peptides with very specific modifications (N-methylation, stapling, etc) can penetrate cell membranes and high affinity and specificity can be achieved but is not straight forward as in antibodies.

The clinical trial https://clinicaltri- als.gov/ct2/show/NCT00093964 mentioned in page 5 is not current.

Connections in Table 1, presumably disulfide bridges, cannot be read.

Author Response

Comments and Suggestions for Authors

In this paper the authors describe peptide therapeutics for the treatment of multiple sclerosis. The review appears to be scientifically sound and of interest in the field since there is little literature on peptide therapeutics used in MS.

In spite of this, several aspects should be improved:

-The review is quite heavy to read and difficult to follow in its current form. A clearer structure is required. Subtitles in each section would greatly help the reader.

We perfectly agree with the reviewer and appreciate his suggestion. Thus, we inserted some subsections throughout the text

-This review reads as a listing of peptides related to MS. Some perspective and trend in the kind of peptides used missing.

We added some perspective in the “Conclusion” section, as required

-A reference to a recent comprehensive review on the topic by Rayatpou and Javan (Pharmacological Research, 2021, 167, 105441) is missing.

The reference was inserted

-Acronyms in the abstract should be explained (MS and EAE).

The Acronyms were explained in the Abstract section

-EAE should be better defined and linked with MS.

We are thankful for the suggestion. Accordingly, we revised the sentence.

-Some explanations should be improved. For instance, the second sentence in the Introduction is confusing and the fragment “macrophages infiltrated the blood-brain barrier (BBB)” is inconsistent.

We modified the sentence to improve the text comprehension.

-In some instances, more formal expressions should be sought. For instance, in page 2, “Till now,” in the third paragraph” should read “Until now,”.

Done

-Peptides can be outstanding therapeutics but not all of them have the properties described in the las paragraph in page 2. Please nuance these properties and state that peptides “may have” them. For instance, it is said peptides have a “small size”, how small? Actually, only a small fraction of peptides with very specific modifications (N-methylation, stapling, etc) can penetrate cell membranes and high affinity and specificity can be achieved but is not straight forward as in antibodies.

We thank the review for his observation.

Regarding peptide properties, our sentence refers to natural peptides having a known role in some physiological mechanisms. Anyway, we replaced “they” with “some of them”.

Actually, peptides size ranges for definition from 2 to 50 amino acid residues, thus we specified this in the text.

-The clinical trial https://clinicaltri- als.gov/ct2/show/NCT00093964 mentioned in page 5 is not current.

Many thanks for this remark. Actually, the studies on cilengitide were completed in 2019. We removed this wrong information from the text

-Connections in Table 1, presumably disulfide bridges, cannot be read.

We fixed the Table 1

Reviewer 3 Report

The manuscript updates a topic of great current interest regarding the potential use of peptides for the treatment of MS.

To make reading more attractive, I have some suggestions for the manuscript.

1) The introduction is very long and with confusing fragments. For example, they start by typing

 "Multiple sclerosis (MS) is a chronic inflammatory disease of the central nervous sys-tem (CNS) that leads to progressive neurodegeneration"

Later they refer:

"MS is a multifaceted disease concerning an intricate interplay of inflammatory cells (T cells, B cells, and macrophages) and CNS resident cells (microglia, astrocytes, oligodendrocytes, and neurons)".

I suggest to facilitate reading, reduce the introduction a bit and open a separate topic to update the most essential of the immunopathogenesis of MS.

2) I suggest that Table 1 briefly include the pharmacological effect of each peptide.

3) Before the conclusions include a new topic to analyze which would be the current difficulties to overcome for the clinical use of peptides against MS. The manuscript only mentions the advantages of peptides but not the existing problems with this technology, such as the need for adequate vehicles for targeted delivery, pharmacokinetic aspects, etc. The models described in this review are based on preclinical results, so a general analysis of what would be necessary for their introduction into humans is necessary.

4) The first paragraph of the conclusions is redundant with what was already said before. I suggest eliminating it because nothing new contributes. Instead, focus on the future prospects of this technology.

5) Carefully review the text to correct various writing errors.

Author Response

Comments and Suggestions for Authors

The manuscript updates a topic of great current interest regarding the potential use of peptides for the treatment of MS.

To make reading more attractive, I have some suggestions for the manuscript.

1) The introduction is very long and with confusing fragments. For example, they start by typing

 "Multiple sclerosis (MS) is a chronic inflammatory disease of the central nervous sys-tem (CNS) that leads to progressive neurodegeneration"

Later they refer:

"MS is a multifaceted disease concerning an intricate interplay of inflammatory cells (T cells, B cells, and macrophages) and CNS resident cells (microglia, astrocytes, oligodendrocytes, and neurons)".

I suggest to facilitate reading, reduce the introduction a bit and open a separate topic to update the most essential of the immunopathogenesis of MS.

We perfectly agree with the reviewer and appreciate his suggestion. Thus, we reduced the introduction section a bit and inserted some subsections throughout the text

2) I suggest that Table 1 briefly include the pharmacological effect of each peptide.

This is a good suggestion but it is difficult to explain in few words the pharmacological effect of each peptide.

3) Before the conclusions include a new topic to analyze which would be the current difficulties to overcome for the clinical use of peptides against MS. The manuscript only mentions the advantages of peptides but not the existing problems with this technology, such as the need for adequate vehicles for targeted delivery, pharmacokinetic aspects, etc. The models described in this review are based on preclinical results, so a general analysis of what would be necessary for their introduction into humans is necessary.

We appreciated this suggestion and decided to insert in the “Conclusion” section some sentences related to potential tools to improve targeted delivery

4) The first paragraph of the conclusions is redundant with what was already said before. I suggest eliminating it because nothing new contributes. Instead, focus on the future prospects of this technology.

In the “Conclusion” section we eliminated some redundant sentences and added some perspective, as required.

5) Carefully review the text to correct various writing errors.

Done

Round 2

Reviewer 3 Report

The authors' responses were adequate and the manuscript was improved according to the recommendations that were made.